# Phytosterols reverse antiretroviral-induced hearing loss, with potential implications for cochlear aging

**Alejandro O. Sodero**[1⊛], **Valeria C. Castagna**[2,3⊛], **Setiembre D. Elorza**[4], **Sara M. Gonzalez-Rodulfo**[1], **María A. Paulazo**[1], **Jimena A. Ballestero**[2], **Mauricio G. Martin**[4‡]*, **María Eugenia Gomez-Casati**[2‡]*

**1** Instituto de Investigaciones Biomédicas, Pontificia Universidad Católica Argentina, Consejo Nacional de Investigaciones Científicas y Técnicas (BIOMED, UCA-CONICET), Buenos Aires, Argentina, **2** Instituto de Farmacología, Facultad de Medicina, Universidad de Buenos Aires, Consejo Nacional de Investigaciones Científicas y Técnicas (CONICET), Buenos Aires, Argentina, **3** Instituto de Investigaciones en Ingeniería Genética y Biología Molecular, Dr. Héctor N. Torres, Consejo Nacional de Investigaciones Científicas y Técnicas (INGEBI-CONICET), Buenos Aires, Argentina, **4** Laboratorio de Neurobiología, Instituto de Investigaciones Médicas Mercedes y Martín Ferreyra, Consejo Nacional de Investigaciones Científicas y Técnicas (INIMEC-CONICET-UNC), Universidad Nacional de Córdoba, Córdoba, Argentina

⊛ These authors contributed equally to this work.
‡ MGM and MEGC also contributed equally to this work.
* mmartin@immf.uncor.edu (MGM); megomezcasati@gmail.com, mgomezcasati@fmed.uba.ar (MEGC)

**Data Availability Statement:** The data supporting the findings of this study are openly available from the Open Science Framework (https://osf.io/xpemk/).

## Abstract

Cholesterol contributes to neuronal membrane integrity, supports membrane protein clustering and function, and facilitates proper signal transduction. Extensive evidence has shown that cholesterol imbalances in the central nervous system occur in aging and in the development of neurodegenerative diseases. In this work, we characterize cholesterol homeostasis in the inner ear of young and aged mice as a new unexplored possibility for the prevention and treatment of hearing loss. Our results show that cholesterol levels in the inner ear are reduced during aging, an effect that is associated with an increased expression of the cholesterol 24-hydroxylase (CYP46A1), the main enzyme responsible for cholesterol turnover in the brain. In addition, we show that pharmacological activation of CYP46A1 with the antiretroviral drug efavirenz reduces the cholesterol content in outer hair cells (OHCs), leading to a decrease in prestin immunolabeling and resulting in an increase in the distortion product otoacoustic emissions (DPOAEs) thresholds. Moreover, dietary supplementation with phytosterols, plant sterols with structure and function similar to cholesterol, was able to rescue the effect of efavirenz administration on the auditory function. Altogether, our findings point towards the importance of cholesterol homeostasis in the inner ear as an innovative therapeutic strategy in preventing and/or delaying hearing loss.

## Introduction

Functional deterioration of the nervous system is frequently observed in the elderly, with a progressive decline in cognitive capacities [1]. Age-related hearing loss (ARHL—presbycusis)

**Funding:** This research was supported by Agencia Nacional de Promoción Científica y Técnica (Argentina) PICT-2018-00539 grant to MEGC and PICT-2018-00648 grant to M.G.M. A.O.S. received financial support from Pontificia Universidad Católica Argentina. The funder had no role in study design, data collection and analysis, decision to publish, or preparation of the manuscript.

**Competing interests:** The authors have declared that no competing interests exist.

**Abbreviations:** ABR, auditory brainstem response; ARHL, age-related hearing loss; CNS, central nervous system; DCN, dorsal cochlear nucleus; DPOAE, distortion product otoacoustic emission; IHC, inner hair cell; OHC, outer hair cell; PBS, phosphate-buffered saline; PFA, paraformaldehyde; PVCN, posteroventral cochlear nucleus; RNS, reactive nitrogen species; ROI, region of interest; ROS, reactive oxygen species; SC, supporting cell; scRNA-seq, single-cell RNA-seq; SPL, sound pressure level.

is a significant problem that leads to a reduced hearing perception, with almost 1 in 3 adults over age 65 experiencing some degree of hearing loss [2,3]. Epidemiologic research has shown that ARHL is strongly related to accelerated cognitive decline and dementia risk in older adults [4–6]. Although the use of hearing aids and/or cochlear implants has been shown to ameliorate many of these linked conditions, ARHL remains considerably undertreated [7]. The high level of noise exposure in modern society makes presbycusis a mixture of acquired auditory stress, trauma, and otological disease superimposed upon an intrinsic aging process [8]. ARHL typically initiates with a loss of synaptic connections between inner hair cells (IHCs) and auditory nerve fibers and a loss of outer hair cells (OHCs) in the high-frequency (basal end) region of the cochlea [3,9–13]. Nonetheless, the cellular and molecular mechanisms underlying these age-related losses remain mostly unknown.

Cholesterol is an important component of neural cell membranes, vital to normal function of the cell and their membrane-associated proteins. Recent studies have shown that cholesterol is abundant in synaptic membranes and modulates a number of protein complexes by directly binding to and/or conditioning the conformation, dynamics, and biophysical properties [14]. Within the central nervous system (CNS), cholesterol plays a key role in synapse formation, cell–cell interactions, and intracellular signaling [15]. Importantly, all cholesterol within the CNS is synthesized in situ due to the fact that peripheral cholesterol cannot cross the blood–brain barrier [16]. Thus, cholesterol levels need to be tightly regulated in the brain, and disruption of cholesterol homeostasis has been linked to cognitive dysfunction and to the development of neurodegenerative diseases [16–20]. Several works have demonstrated a reduction in cholesterol levels in the hippocampus during aging [21,22], leading to a profound effect on the plasma membrane structure and cell function [15,19,23–26]. Reduced cholesterol content in the aged brain and in neurodegenerative diseases was also reported in humans' brain samples [21,27–34]. Among the causes for age-associated cholesterol loss, an increase in the levels of the cholesterol-hydroxylating enzyme CYP46A1 in the hippocampus [24], associated with oxidative stress accumulation, has been proposed [22,35]. CYP46A1 is a brain-specific enzyme up to now reported in pyramidal cells and interneurons of the cortex and hippocampus and in the Purkinje cells of the cerebellum [36–38]. It has been shown that a decrease in the synaptic cholesterol content during aging in the mouse hippocampus, mainly due the CYP46A1 up-regulation, interferes with synaptic receptor mobility and endocytosis and leads to memory impairments [18]. Interestingly, restoring brain cholesterol levels can rescue biochemical, synaptic, and cognitive deficits of aged mice [19,39] and in a mouse model of Huntington's disease [40].

OHCs are sensory cells implicated in the mechanical amplification of sounds and in the frequency selectivity in mammals [41]. OHCs respond to changes in membrane voltage with changes in length, a phenomenon known as electromotility, which is brought about by the motor protein prestin, a polytopic integral membrane protein present in the OHC's lateral wall [42,43]. The organization of the OHC lateral wall is exclusive among hair cells and other mammalian cell types because it contains a tightly regulated level of cholesterol [44–46]. It has been postulated that alterations in the cholesterol content in the OHC lateral wall might modulate the function and/or distribution of prestin within the plasma membrane [45,47].

All of these evidences led us to hypothesize that cholesterol deficiency might play a role in the associated cochlear pathologies that occur during aging. However, the role of cholesterol homeostasis in the physiopathology of the inner ear has not been studied [48]. The aim of the present study was to test this hypothesis by analyzing CYP46A1 levels and cholesterol content in the inner ear of young and aged mice. Moreover, we tested the consequences of pharmacologically altering the cholesterol content in vivo with the activation of CYP46A1 with efavirenz [49,50], an FDA-approved anti-HIV drug, on the auditory function. The effect of dietary supplementation with phytosterols was also tested in the inner ear of mice treated with efavirenz.

Our findings show for the first time the importance of cholesterol homeostasis in the inner ear as a pharmacotherapeutic strategy to prevent and/or delay hearing loss.

## Results

### Cholesterol homeostasis in the inner ear during aging

Cholesterol hydroxylation into 24(S)-hydroxycholesterol by CYP46A1, and elimination of this oxysterol through the blood–brain barrier, is the main mechanism for cholesterol removal from the brain [51]. Thus, to test the possibility that cholesterol changes would be involved in ARHL, we analyzed the levels of the catabolic enzyme CYP46A1 in the organ of Corti of young and aged C57BL/6J mice. C57BL/6J is the mouse strain most widely used for aging studies due to its longevity, up to 27 to 29 months. Previous work in this strain has shown that aging increases CYP46A1 levels in the hippocampus of 24-month-old mice, leading to cholesterol loss and cognitive decline [19]. However, these deficits are not evident until the mice reach 19 months of age. Thus, we decided to explore the CYP46A1 levels at an advanced age in the inner ear of C57BL/6J mice. Cochleae were harvested from 2-month-old and 24-month-old C57BL/6J mice and fixed for histological analysis. Organ of Corti whole mounts were immunostained with antibodies against CYP46A1 and calretinin to label cochlear hair cells (Fig 1A). As seen from the representative confocal pictures, CYP46A1 labeling was observed in both sensory and supporting cells (SCs) in young and aged mice (Fig 1A). Whole mount immunostainings (Fig 1A) and quantitative analysis at different cochlear locations: apical, medial, and basal (Fig 1B) showed a significant increase in CYP46A1 fluorescence intensity in the sensory epithelia of aged mice. In IHCs, there was an increase in CYP46A1 expression at the 3 regions of the cochlea (t test, $p < 0.0001$ at the apical/medial and $p < 0.01$ at the basal end). In those surviving OHCs in 24-month-old mice, we also found an increase in CYP46A1 fluorescence intensity along the whole cochlea (t test, $p < 0.001$ at the apical and $p < 0.0001$ at the medial/ basal region). In the SCs' area, we quantified the CYP46A1 fluorescence intensity by counting all the positive-labeled cells surrounding both IHCs and OHCs. We found a significant increase in CYP46A1 expression in SCs of 24-month-old mice, but it was statistically significant only in the medial and basal region of the cochlea (t test, $p < 0.001$ at the basal and $p < 0.0001$ at the medial region) (Fig 1B). According to previous results showing that OHCs in the C57BL/6J strain degenerate relatively early in life [52–54], an important loss of approximately 70% of OHCs could be observed at 2 years of age as visualized by the calretinin staining (Figs 1A, S1A, and S1B). OHCs were counted along the whole cochlear sections combining the calretinin-positive signal together with light microscopy, based on the assessment of all present and absent hair cells in all the sections. Representative pictures in S1A Fig show that at 2 years of age, there were also changes in IHCs morphology together with some IHCs loss, indicated by white arrowheads. In young mice, calretinin staining was mainly observed in IHCs, and there was a weak labeling in OHCs (Figs 1A and S1A). Similarly, single-cell RNA-seq (scRNA-seq) studies from adult CBA/J mouse cochleae have shown a weak calretinin expression in OHCs compared to IHCs [55]. Notably, we observed an increase in the calretinin labeling in those surviving OHCs in mice at 2 years of age (Figs 1A and S1A).

It has been previously shown that cholesterol content is a critical player in the modulation of hearing in mice due to its important role in the proper distribution of prestin within the lateral plasma membrane of OHCs [45,47]. To analyze if the observed increase in CYP46A1 levels in aged mice leads to a reduction in cholesterol content in the sensory epithelium, we used the fluorescent antibiotic filipin to label cholesterol and examine the levels of this sterol in cochlear tissue. In accordance with published findings [45], cholesterol distribution in OHCs from young adult mice showed a strong intracellular filipin staining surrounded by a region of

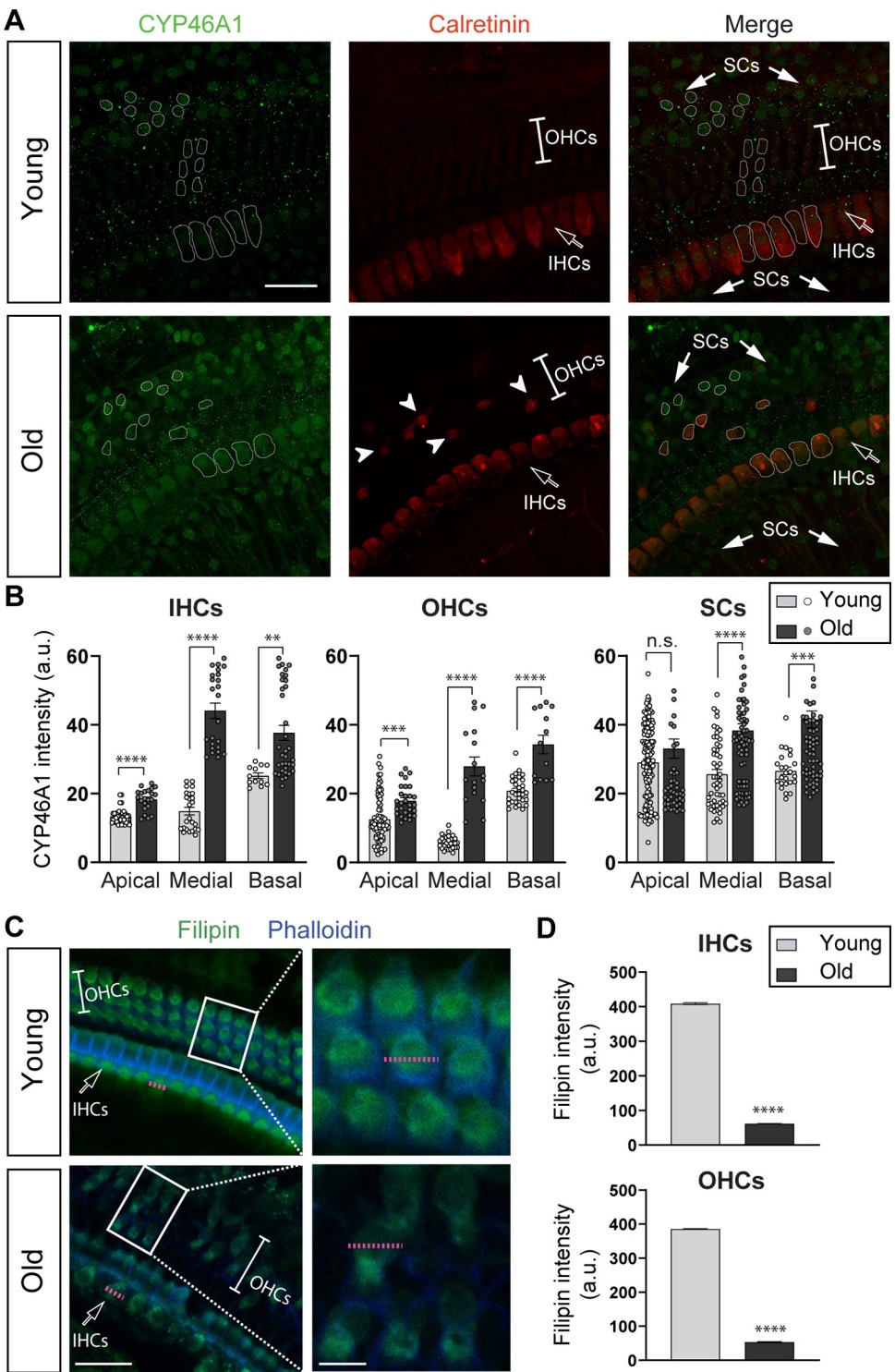

**Fig 1. CYP46A1 expression and cholesterol content in aged mouse cochlea.** (**A**) Representative confocal images in the xy projection of whole mounts organ of Corti from the medial region immunolabeled for CYP46A1 (green) and calretinin (red) from young (top) and old (bottom) C57BL/6J mice (scale bar = 30 μm). Filled arrows indicate the SCs' area. Arrowheads point to surviving OHCs in aged mice. IHCs and OHCs' area are also indicated. (**B**) Quantitative data obtained from young (*n* = 5) and aged (*n* = 6) mice. CYP46A1 fluorescence intensity (a.u., arbitrary units) was measured in IHCs (left), OHCs (middle), and SCs (right) at 2 months and 2 years of age at 3 regions of the cochlea: apical, medial, and basal end (Young: *n* = 43 IHCs, 78 OHCs, and 144 SCs at the apical; 25 IHCs, 29 OHCs, and 51 SCs

at the medial; and 12 IHCs, 31 OHCs, and 24 SCs at the basal region. Old: $n = 22$ IHCs, 28 OHCs, and 55 SCs at the apical; 24 IHCs, 17 OHCs, and 79 SCs at the medial; and 28 IHCs, 13 OHCs, and 69 SCs at the basal region). Examples of ROIs that correspond to each cellular type (IHC, OHC, and SC) were drawn. In aged mice, there was a significant increase in CYP46A1 immunolabeling in IHCs, OHCs, and SCs compared to young ears at all the cochlear regions (except at the apical end in SCs). n.s. = not significant. Asterisks represent the statistical significance ($t$ test, ** = $p < 0.01$, *** = $p < 0.001$ **** = $p < 0.0001$). (**C**) Representative confocal images of whole mounts organ of Corti from the medial region of the cochlea stained for filipin to label cholesterol (green) and phalloidin for visualizing actin in stereocilia (blue) from young (top) and old (bottom) C57BL/J mice (scale bar = 30 μm). Rectangles indicate the region of the zoomed-in areas shown on the right using a 63× objective (scale bar = 8 μm). Pink dotted lines represent the diameter of individual IHC and OHC. (**D**) Filipin staining quantified with a pixel intensity graph (a.u., arbitrary units) along a line (shown as dotted pink lines in panel **C**) crossing either individual OHCs or IHC in young ($n = 3$) and aged ($n = 6$) mice at all the regions of the cochlea. In aged mice, there was a significant reduction in cholesterol content in IHCs (top panel) and surviving OHCs (bottom panel) compared to young ears (Young: $n = 104$ IHCs and 200 OHCs; Old: $n = 52$ IHCs and 22 OHCs). Asterisks represent the statistical significance ($t$ test, **** = $p < 0.0001$). The data underlying this figure can be found at https://osf.io/xpemk/. IHC, inner hair cell; OHC, outer hair cell; ROI, region of interest; SC, supporting cell.

lower staining corresponding to the lateral wall membrane of the cell, as indicated by a dashed pink line in an individual OHC in the magnified inset in Fig 1C. As seen in the representative confocal pictures (Fig 1C) and in the pixel intensity graph (Fig 1D, bottom panel), there was a significant reduction of the filipin staining in OHCs at 2 years of age compared with younger mice ($t$ test, $p < 0.0001$). Although aged cochleae showed an evident OHCs loss, filipin staining was quantified along the diameter of surviving OHCs to unequivocally determine the cholesterol content per cell, as indicated by the dashed pink line in the magnified inset in Fig 1C, bottom panel. The filipin labeling in IHCs also showed a significant reduction in pixel intensity in aged cochleae (Fig 1D, upper panel, $t$ test, $p < 0.0001$).

## Pharmacological activation of CYP46A1 impairs the function of OHCs in vivo

The above results indicate that cholesterol content decreases in the inner ear during aging and that this cholesterol decrease correlates with an enhanced staining of the cholesterol hydroxylating enzyme CYP46A1. Thus, we postulated that CYP46A1 activation might reduce the cholesterol content in young mice, leading to hearing deficits. To test this hypothesis, we used the anti-HIV medication efavirenz as it is known that it enhances CYP46A1 activity [49,50] and then assessed the auditory function. It has been demonstrated that efavirenz is able to cross the blood–brain barrier and stimulates CYP46A1 activity in mice at doses 300 times lower than those used for HIV patients [49]. We administered efavirenz at a dose of 0.09 mg/kg/day for 4 weeks in the drinking water of C57BL/6J mice at 2 months of age. It has been described that 2-month-old C57BL/6J mice have a good hearing sensitivity and auditory threshold elevations in this strain starts at 8 to 10 months of age [56]. Two complementary tests were used to evaluate cochlear function that allow differential diagnosis of OHC versus IHC/auditory nerve dysfunction throughout the cochlea, from the low frequency apical turn to the high-frequency basal end. The first test was the auditory brainstem responses (ABRs) that are sound-evoked potentials produced by neuronal circuits in the ascending auditory pathway measured from scalp electrodes. ABR waveforms comprise a succession of peaks within the first approximately 7 ms after sound stimulus onset (Peaks 1 to 5). The first peak represents the summed sound-evoked spike activity at the first synapse between IHCs and afferent nerve fibers [57]. The second complementary measurement of cochlear output was to assess OHCs function through distortion product otoacoustic emissions (DPOAEs) recorded from the external auditory canal [58]. In the normal cochlea stimulated simultaneously by 2 close pure tone frequencies, distortions are created by nonlinearities in OHC transduction. These distortion products are

amplified by OHCs electromotility, causing motion at the distortion frequencies that propagates to the middle ear and can be detected by a microphone in the external ear canal [58,59]. Efavirenz treatment produced a slight increase, although not statistically significant, in ABR thresholds compared to control untreated mice at either 8 or 12 weeks, the ages of the beginning and end of the treatments, respectively (Fig 2A and 2B) (Kruskal–Wallis test: df = 2, $p > 0.05$ at all the frequencies). There were no differences in ABR thresholds between controls at 8 and 12 weeks (Fig 2B) (Mann–Whitney $u$ test: df = 1, $p > 0.05$ at all the frequencies). Suprathreshold ABR peak 1 amplitudes were not modified after efavirenz administration, indicating no drug-related changes in cochlear neural function (Fig 2C) (Kruskal–Wallis test: df = 2, $p > 0.05$ at all the frequencies). However, we observed a 20- to 30-dB SPL elevation of DPOAEs thresholds at the mid-frequency cochlear region in mice treated with efavirenz compared to controls (Fig 2D) (Kruskal–Wallis test: df = 2, $p = 0.02$, Dunn's post-tests, $p < 0.05$ at 11.33,16, and 22.65 kHz), indicating that OHC function is degraded after efavirenz treatment. The mean data for the emission amplitude-versus-level functions for the DPOAEs at 16 kHz was reduced after efavirenz treatment (Fig 2E). Fig 2F shows examples of fast Fourier transforms of the DPOAE in the frequency domain at baseline, with f1 and f2 being 13.34 and 16 kHz, respectively, at 80 dB SPL in control and treated mice. The cubic DPOAE (2f1-f2) was diminished in mice treated with efavirenz compared to controls (Fig 2F).

After physiological testing, cochleae were harvested and fixed to determine cholesterol levels in OHCs using filipin and phalloidin to label the hair cells' stereocilia. Filipin staining showed a reduction in cholesterol levels in OHCs in the group treated with efavirenz compared to untreated controls (Fig 3A, upper and central panels). The filipin staining pattern in OHCs was certainly evident in the pixel intensity graph, which plots pixel intensities along the width of 2 consecutive OHCs (indicated by a dashed line in Fig 3A) from each of the 3 rows at the basal region of the cochlea as a representative example (Fig 3B). After efavirenz treatment, there was a significant reduction in the cholesterol content in OHCs along the 3 regions of the cochlea, from the basal to the apical end, as shown in the relative intensity graph in Fig 3C (one-way ANOVA, df = 2, Tukey's test, $p < 0.0001$ for all the comparisons at all the regions). Quantification analysis of filipin staining in IHCs and SCs also showed a reduction in the cholesterol content in these cells at the 3 regions of the cochlea (one-way ANOVA, df = 2, Tukey's test, $p < 0.0001$ for IHCs at all the regions and $p < 0.001$ and $p < 0.0001$ for SCs at the medial and apical/basal regions, respectively) (S2A–S2C Fig). As expected, activation of CYP46A1 by efavirenz in OHCs, IHCs, and SCs (Fig 1A and 1B) led to a decrease in cholesterol levels in these 3 cell populations throughout the cochlea (Figs 3 and S2). At the end of the treatment, we did not observe any hair cell loss by histological examination of the whole mounts organ of Corti using antibodies against phalloidin that label stereocilia of sensory hair cells (Fig 3A) and calretinin/prestin (Fig 4A) that label both IHCs and OHCs, respectively. Our results clearly show that efavirenz treatment is able to stimulate CYP46A1 activity in the sensory epithelium leading to cholesterol loss in OHCs and DPOAEs threshold elevations, suggesting an impaired OHCs function.

## Impaired OHC function in efavirenz-treated mice can be rescued by phytosterols

Phytosterols are plant naturally occurring compounds structurally and functionally similar to cholesterol in mammals [60]. In contrast to circulating cholesterol, dietary plant sterol-esters can cross the blood–brain barrier and accumulate in the membranes of CNS cells [61]. It has recently been shown that a diet enriched with plant sterols can prevent memory impairments induced by cholesterol loss in senescence-accelerated SAMP8 mice [62]. We reasoned that a

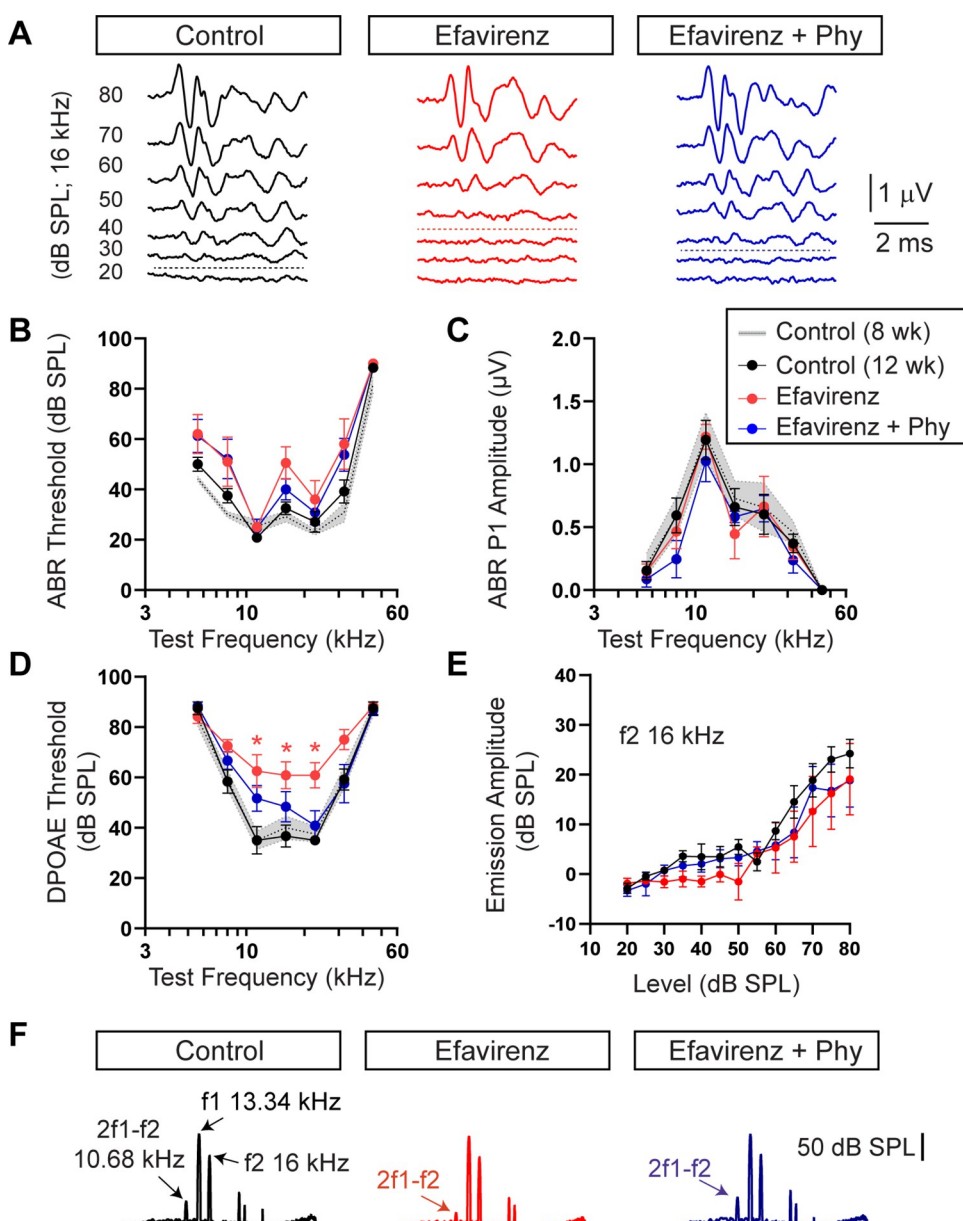

**Fig 2. Auditory function in control, efavirenz, and efavirenz plus phytosterols-treated mice. (A)** Representative ABR waveforms from control (black), efavirenz-treated (red), and efavirenz-treated plus Phy (blue) mice recorded at 16 kHz tone bursts at increasing sound pressure levels (dB SPL). Thresholds are indicated by dashed traces and were determined by the presence of peak 1. Scale bar applies to the 3 recordings. ABR thresholds showed a slight increase in those mice treated with efavirenz that partially recovered when supplemented with Phy. Mean ABR thresholds (**B**) and ABR peak 1 (P1) amplitudes at 80 dB SPL (**C**) for control ($n = 6$), treated with efavirenz ($n = 6$), and treated with efavirenz together with Phy ($n = 6$) mice at different test frequencies. We included controls at 12 weeks; the age after the treatment (black; $n = 6$) and controls at 8 weeks, before starting the treatment (gray; $n = 6$). (**D**) DPOAEs thresholds for control (12 weeks; black; $n = 6$ and 8 weeks; gray; $n = 6$), treated with efavirenz ($n = 6$), and treated with efavirenz together with Phy ($n = 6$) mice at different test frequencies. DPOAEs thresholds showed a significant increase in those mice treated with efavirenz that partially recovered when supplemented with Phy. (**E**) Mean DPOAEs amplitudes versus level functions with f2 = 16 kHz in the 3 groups of mice. Group means ± SEM are shown (**B-E**). (**F**) Examples of fast Fourier transform traces showing the cubic DPOAE amplitude (2f1-f2), at f1 = 13.34 kHz; f2 = 16 kHz with f2 input level of 80 dB SPL. Scale bar of 50 dB SPL applies to the 3 recordings. Asterisks represent the statistical significance (Kruskal–Wallis test, followed by Dunn's post-tests * = $p < 0.05$). The data underlying this figure can be found at https://osf.io/xpemk/. ABR, auditory brainstem response; Phy, phytosterols; SPL, sound pressure level.

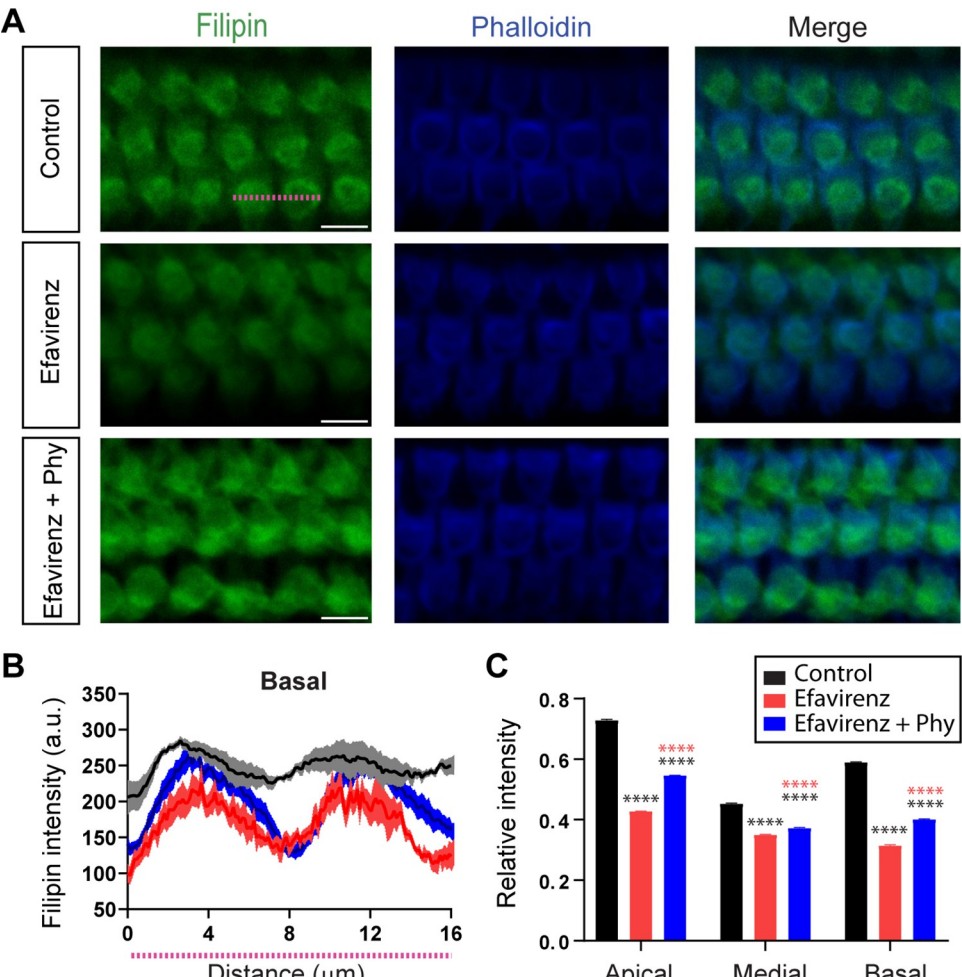

**Fig 3. Cholesterol content in OHCs from control, efavirenz, and efavirenz plus phytosterols-treated mice.** (**A**) Representative confocal images of whole mounts organ of Corti from the OHCs' area of the basal region stained with phalloidin (blue; to visualize actin in stereocilia) and filipin (green; to label cholesterol) in control, treated with efavirenz, and treated with efavirenz plus Phy mice. Pink line represents approximately 16 μm, i.e., the width of 2 consecutive OHCs (scale bars = 8 μm). (**B**) Pixel intensities plot obtained from all the sections (Control: $n$ = 14 cells from 6 animals; efavirenz: $n$ = 12 cells from 6 animals and efavirenz together with phytosterols: $n$ = 28 cells from 6 animals) along the width of 2 consecutive OHCs from each of the 3 rows at the basal region of the cochlea shows that after efavirenz treatment, there was a reduction in cholesterol and a slight increase in filipin staining when supplemented with Phy. Pink line underneath the x-axis represents the width of 2 consecutive OHCs (**C**) Relative intensity bar graph showing the changes in filipin staining in the 3 groups of mice at the 3 regions of the cochlea (apical, medial, and basal). There was a significant reduction in cholesterol levels after efavirenz treatment. Treatment with efavirenz together with Phy showed an increase in filipin staining at the 3 regions compared with mice treated with efavirenz alone. However, when compared to controls, there was a significant reduction in filipin staining at the 3 regions of the cochlea (Control: $n$ = 80 OHCs at the apical; 140 OHCs at the medial; and 70 OHCs at the basal region from 6 mice; efavirenz: $n$ = 200 OHCs at the apical; 70 OHCs at the medial; and 65 OHCs at the basal region from 6 mice and efavirenz together with phytosterols: $n$ = 92 OHCs at the apical; 115 OHCs at the medial; and 77 OHCs at the basal region from 6 mice). Group means ± SEM are shown. Asterisks represent the statistical significance (one-way ANOVA, followed by Tukey's test, **** = $p$ < 0.0001). The data underlying this figure can be found at https://osf.io/xpemk/. OHC, outer hair cell; Phy, phytosterols.

diet enriched with phytosterols could rescue the cholesterol loss–dependent inner ear defects triggered by efavirenz administration. To test this hypothesis, an experimental group was supplemented with phytosterols in the mice's food for 3 weeks (ad libitum) in addition to the efavirenz treatment.

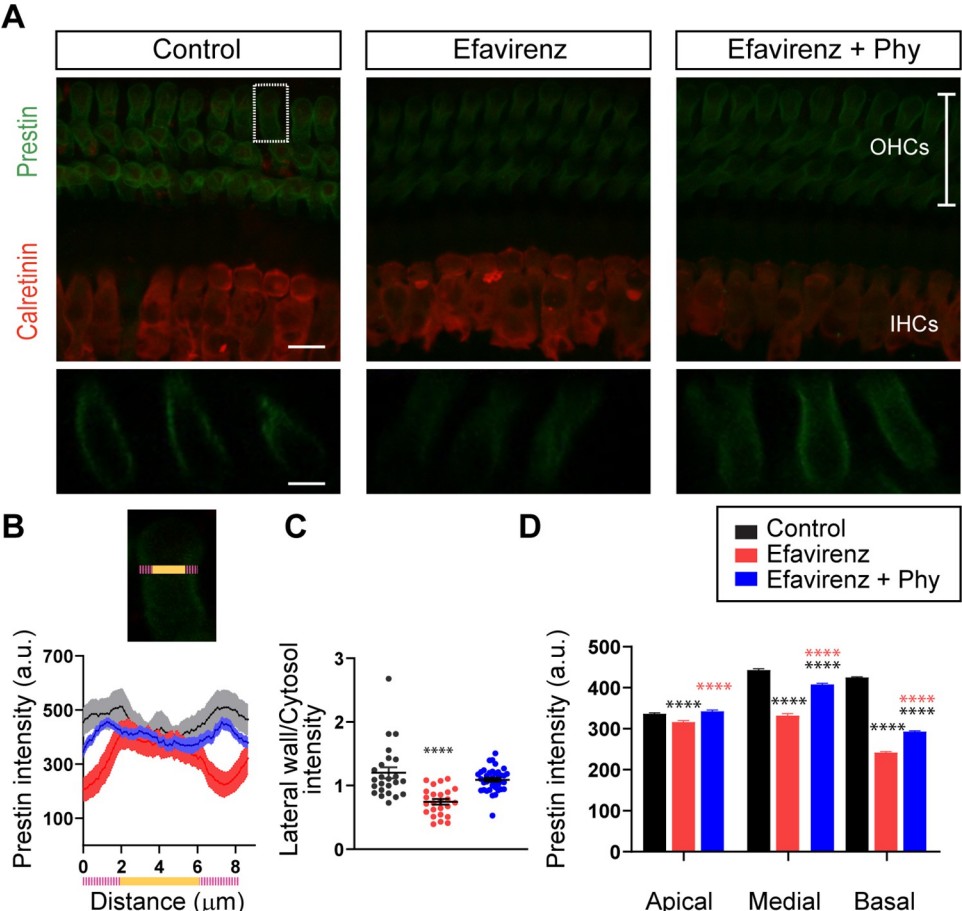

**Fig 4. Prestin expression in OHCs from control, efavirenz, and efavirenz plus phytosterols-treated mice.** (**A**) Representative confocal images of whole mounts organ of Corti from the medial region immunolabeled for calretinin (red) and prestin (green) in control, treated with efavirenz, and treated with efavirenz plus Phy mice (top) (scale bar = 16 μm). Cochlear whole mount yz projection of 3 OHCs from each row (bottom) (scale bar = 8 μm). (**B**) Mean pixel intensity plot along the width of one individual OHC from each of the 3 rows at the medial region of the cochlea as a function of distance in the 3 groups of mice (bottom panel). High-magnification image of one OHC from the selected area indicated by a white rectangle on the confocal image in **A** (top panel). The pink and orange line represent approximately 8 μm, the width of one OHC (pink ends indicate the region corresponding to the lateral wall, and orange the cytoplasm of the cell). (**C**) Ratio between the average fluorescence intensity in the lateral wall and the cytoplasm to visualize the change in distribution of prestin within each OHC at the medial region of the cochlea (Control: *n* = 24 cells from 6 mice; efavirenz: *n* = 25 cells from 6 mice and efavirenz together with phytosterols: *n* = 38 cells from 6 mice). (**D**) Bar graph showing the mean in the fluorescence intensity of prestin at the 3 cochlear regions (apical, medial, and basal). The intensity was measured by tracing a line through 3 consecutive OHCs. (Control: *n* = 110 OHCs at the apical, 60 OHCs at the medial, and 76 OHCs at the basal region; efavirenz: *n* = 75 OHCs at the apical, 90 OHCs at the medial, and 60 OHCs at the basal region and efavirenz together with phytosterols: *n* = 110 OHCs at the apical, 105 OHCs at the medial, and 80 OHCs at the basal region). After efavirenz treatment, there was a reduction in prestin labeling in the OHC lateral wall. Supplementation with Phy increased the presence of prestin in the OHCs membrane compared with mice following efavirenz treatment alone. Group means ± SEM are shown. Asterisks represent the statistical significance (one-way ANOVA, followed by Tukey's test, **** = *p* < 0.0001). The data underlying this figure can be found at https://osf.io/xpemk/. OHC, outer hair cell; Phy, phytosterols.

As shown in Fig 2A–2C, there were no changes in ABR thresholds nor suprathreshold ABR peak 1 amplitudes in mice treated with efavirenz plus phytosterols compared to control or efavirenz-treated animals (Kruskal–Wallis test: df = 2, *p* > 0.05 at all the frequencies). However, there was a decrease, although not statistically significant, in DPOAEs thresholds in mice treated with efavirenz together with phytosterols compared to the ones treated with efavirenz

alone (Fig 2D–2F). Importantly, there was no statistical difference in DPOAEs thresholds in mice treated with efavirenz plus phytosterols compared to controls (Kruskal–Wallis test, df = 2, $p > 0.05$ at all the frequencies).

To evaluate whether phytosterols can restore the sterol content in the OHC membrane, cochleae were dissected and stained with filipin. Previous reports have shown that filipin is able to recognize phytosterols in eukaryotic membranes [63]. Representative confocal microscopy images of whole mounts of the organ of Corti stained with filipin and phalloidin to label hair cells' stereocilia are shown in Fig 3A. The pixel intensity graph along the width of 2 consecutive OHCs from the basal region showed the increase in filipin intensity in the group treated with phytosterols compared with mice treated with efavirenz alone (Fig 3B). The same increase was observed at the apical, medial, and basal regions of the cochlea (one-way ANOVA, df = 2, Tukey's test, $p < 0.0001$ at all the regions) (Fig 3C). Although sterol levels did not reach those measured in the controls (one-way ANOVA, df = 2, Tukey's test, $p < 0.0001$ at all the regions), the increase in filipin staining in the efavirenz plus phytosterol-treated group indicates that phytosterols can partially restore the sterol content in the OHCs membrane (Fig 3C). Quantification analysis of phalloidin fluorescence intensity in OHCs showed no difference among the 3 groups of mice. Control: apical 1,023 ± 13.13; medial 1,364 ± 8.98; and basal 1,289 ± 11.79. Efavirenz: apical 1,027 ± 7.21; medial 1,355 ± 8.70; and basal 1,297 ± 11.83. Efavirenz + Phy: apical 1,020 ± 6.04; medial 1,369 ± 17.77; and basal 1,265 ± 8.43. (Arbitrary units mean ± SEM, one-way ANOVA, df = 2, $p > 0.05$). Altogether, these results suggest that phytosterols' supplementation in the diet partially rescues the OHC function in efavirenz-treated mice.

## Altered distribution of prestin in the OHCs' membrane triggered by efavirenz can be rescued by phytosterols

Taking into account that prestin is a critical component of the OHCs lateral membrane and is essential for their electromotility, we wondered if elevation in DPOAEs thresholds by reduction in the cholesterol content could be explained by alterations in the distribution of prestin within the plasma membrane. Prestin levels in OHCs from control, efavirenz-treated, and efavirenz plus phytosterols-treated mice were evaluated by immunohistochemistry using antibodies against prestin and calretinin to visualize hair cells. As seen in the representative confocal images of whole mounts of the organ of Corti from the medial region (Fig 4A), there was a reduction in prestin immunolabeling in ears treated with efavirenz compared to controls. After rotating the z-stack image to view the projection of 3 OHCs from each row as a cross section through the cochlear epithelium, the reduction in prestin labeling in the lateral wall was even clearer in mice treated with efavirenz (i.e., the yz plane; Fig 4A, bottom panel). As shown in Fig 4A–4C, in control mice, prestin was concentrated in the lateral region of the OHCs' membrane. After efavirenz treatment, prestin labeling was reduced in the lateral wall and distributed along the cell's body. Notably, phytosterols supplementation in the diet increased prestin levels in the OHCs membrane compared with efavirenz-treated mice (Fig 4A–4C), leading to a partial recovery of the normal prestin distribution. Fig 4B on the bottom panel shows a representative plot of prestin intensity at the medial region of the cochlea along the width of one OHC from each row from all the sections that were imaged at the confocal microscope from the 3 groups of mice. The top panel on Fig 4B shows a magnification image of an individual OHC of the selected area indicated by a white rectangle on Fig 4A as an example of how we draw the line along the OHC to perform the prestin intensity analysis as a function of the distance of the cell. The pink and orange colored line represents 8 μm, the average diameter of a single OHC (Fig 4B). Pixel intensity values between the region corresponding to 0 to 2 μm and 6 to 8 μm represent the approximate width of the lateral wall of one individual OHC

(indicated by the pink color of the line) and the orange part of the line indicates the cytoplasm of the OHC (Fig 4B). Prestin intensity plot in the areas corresponding to the lateral wall (indicated with pink color) showed a reduction in pixel intensity in the efavirenz-treated group compared to controls and a slight, but significant, increase in the prestin signal after phytosterols supplementation. To quantify the change in the distribution of prestin within each OHC at the medial region of the cochlea, we computed the average fluorescence in the membrane area (Distances 0 to 2 and 6 to 8 μm, indicated by the pink line in Fig 4B) and in the cytoplasm (Distances 2 to 6 μm, orange line in Fig 4B) and calculated the ratio between the membrane and the cytoplasm fluorescence. As shown in Fig 4C, in the control condition, the ratio rendered a value of $1.20 \pm 0.09$, indicating a higher presence of prestin in the cell membrane compared to the cytoplasm. In efavirenz-treated animals, this ratio dropped significantly to $0.74 \pm 0.04$ (one-way ANOVA, df = 2, Tukey's test, $p < 0.0001$ at all the regions) showing a higher presence of prestin in the cytoplasm compared to the membrane. This effect was reverted in the animals treated with phytosterols in which the ratio was $1.09 \pm 0.03$. This ratio was not significantly different compared to control mice (one-way ANOVA, df = 2, $p > 0.05$). Finally, we computed the mean prestin fluorescence intensity at all the regions of the cochlea (apical, medial, and basal) in the 3 groups of mice by drawing a line through the middle of 3 consecutive OHCs (Fig 4D). The reduction in prestin staining with efavirenz was evident at all regions of the cochlea compared to control mice (one-way ANOVA, df = 2, Tukey's test, $p < 0.0001$ at all the regions). Importantly, addition of phytosterols to the diet produces a significant increase in the pixel intensity at the 3 regions compared with mice treated with efavirenz alone (one-way ANOVA, df = 2, Tukey's test, $p < 0.0001$ at all the regions). Although the increase in prestin immunostaining after addition of phytosterols did not reach the control levels at the medial and basal region of the cochlea, there was a significant increase compared with mice treated with efavirenz alone. Altogether, these data indicate that cholesterol content in OHCs decrease with the activation of CYP46A1 with efavirenz leading to a reduction of the motor protein prestin in the plasma membrane resulting in elevations in DPOAEs thresholds. Notably, we could partially restore the OHCs' function by phytosterols supplementation.

## Discussion

Mammalian hearing function depends on the active motility of OHCs to amplify the basilar membrane motion [42,62]. OHCs are highly polarized cells divided into 3 domains: its apical pole with the cuticular plate and stereocilia, its basal pole containing the nucleus and auditory synapses, and the lateral wall. The OHC lateral wall has a unique trilaminate organization above the nuclear level and it has been demonstrated that OHC electromotility resides in its lateral wall [64–67]. Thus, the lipid composition of the lateral wall membrane is critical to cochlear mechanics [68]. According to this idea, [68] described how membrane cholesterol modulates cochlear electromechanics, and [45] showed a dynamic and reversible relationship between membrane cholesterol levels and voltage dependence of prestin-associated charge movement in OHCs, tuning the function of these cells.

Previous works have shown that aging is associated with alterations in cholesterol metabolism in the CNS, mainly in the brain, which is dependent on the region studied. In the hippocampus, aging triggers a mild but consistent cholesterol loss from neurons due to the enhanced activity of the cholesterol 24-hydroxylase or CYP46A1, the enzyme that catalyzes the conversion of cholesterol to 24S-hydroxycholesterol [24]. Cholesterol loss in aged hippocampal neurons affects postsynaptic AMPA receptors lateral mobility, impairs receptor endocytosis, and also impacts on the signaling pathways activated by neuronal stimulation. All these events result in cognitive deterioration in old individuals. Interestingly, these cognitive defects

could be rescued either by cholesterol perfusion in the lateral ventricle in old mice [19] or by pharmacological CYP46A1 inhibition in vivo [39]. This evidence led us to investigate if an altered cholesterol metabolism in the inner ear would be an underlying event associated with ARHL [48].

We found for the first time by immunofluorescence that CYP46A1 is expressed in the inner ear in 3 different populations of cells (IHCs, SCs, and OHCs) in young C57BL/6J mice. Interestingly, CYP46A1 levels are significantly increased in 24-month-old mice in these 3 populations. Paralleling the CYP46A1 increase, we found a significant reduction in the cholesterol levels in sensory cells as identified by filipin staining of whole mounts organ of Corti in 24-month-old mice. To test if there is a direct cause–effect relationship among CYP46A1 activity, cholesterol loss and the auditory function, we treated 2-month-old mice with efavirenz, an anti-HIV drug able to specifically activate CYP46A1. We choose to start the treatment at 2 months to avoid the auditory threshold elevations with age developed by the C57BL/6J strain, which starts at 8 to 10 months of age [56]. Supporting our hypothesis, we found that efavirenz treatment led to a reduction in the cholesterol levels of OHCs that was accompanied by an altered OHC function. Indeed, otoacoustic emissions were measured as an objective indicator of active cochlear amplification through electromechanical properties of OHCs [69,70]. We observed a 20- to 30-dB SPL elevation of DPOAEs thresholds at the mid-basal frequency cochlear region in mice treated with efavirenz compared to controls indicating that OHC function is degraded when cholesterol levels are reduced in these cells. A mild, but not statistically significant, ABR threshold elevation was observed after efavirenz treatment, alteration that can be explained by the deficits in OHC function due to cholesterol loss. Notably, suprathreshold ABR peak 1 amplitudes showed no changes at all the frequencies tested indicating that cochlear neural function was not altered by the treatment.

Then, we tested the possibility that altered OHC function due to cholesterol loss could be rescued by dietary supplementation of phytosterols to efavirenz-treated mice. We analyzed the effect of phytosterols based on previous evidence showing that cognitive impairments due to cholesterol loss in hippocampal neurons in an aging mouse model were rescued by dietary phytosterol supplementation [62]. We found an increase in filipin staining in OHCs of whole mounts organ of Corti after 3 weeks of phytosterols' supplementation in the diet, indicating that they might reach the OHCs membrane. Accordingly, we found a reduction in DPOAEs thresholds in mice treated with efavirenz plus phytosterols, which were statistically indistinguishable from untreated controls, suggesting that phytosterols supplementation in the diet partially restored OHCs' function.

Looking for the possible cause by which cholesterol reduction in efavirenz-treated cells would lead to OHC dysfunction, we assessed the prestin levels in OHCs of mice from the control, efavirenz, and efavirenz plus phytosterols groups by immunostaining. Prior studies have shown the critical role of cholesterol to tune the membrane-based motor to function at maximal gain in the OHC receptor potential range [45]. Prestin was uniformly distributed along the OHC lateral plasma membrane in control cells and reduced levels of prestin together with a miss-distribution of this protein along the cell was observed in OHCs from low-cholesterol efavirenz-treated mice. Strikingly, phytosterols supplementation increased the levels of prestin in the OHCs with an apparent recovery of the normal prestin distribution in the lateral wall. Prestin alterations would probably not be the only defect that results from reduced cholesterol in OHCs. It has been described that the composition of membrane lipids considerably affects the biophysical and mechanical properties of the plasma membrane such as the fluidity and stiffness [46], which can modulate the gating of voltage-gated channels [71]. Indeed, it has been reported that cholesterol influences voltage-gated calcium and BK-type potassium channels in mature afferently innervated hair cells from chicks [72]. Additionally, cholesterol may

exert its effect by directly binding to other membrane proteins affecting their conformation and dynamics [73,74]. Thus, the possible influence of cholesterol on channel biophysics, as well as protein localization in the membrane, indicates that cholesterol is a key factor in auditory physiology. In any case, our results clearly reveal that a reduced cholesterol content in OHCs due to CYP46A1 activation leads to a reduction of the levels and miss-distribution of the motor protein prestin, which is paralleled by the elevation of DPOAEs thresholds. We propose that this phenomenon may occur during the early stages of aging, before OHCs' degeneration, and would be at least one of the multiple causes triggering ARHL/presbycusis.

ARHL is a multifactorial progressive disease produced by different factors [8]. Damage to peripheral sensory and neural elements as well as changes occurring along the central auditory pathway contribute to age-related decline in acuity [75–77]. Peripheral degeneration involves the degeneration of the *stria vascularis*, sensory hair cells, spiral ganglion neurons, and fibrocytes. Accumulation of reactive oxygen and reactive nitrogen species (ROS and RNS, respectively) as well as intracellular calcium homeostasis have been proposed as contributory factors making OHCs, especially those in the high-frequency region of the cochlea, most vulnerable to aging [78,79]. Indeed, by immunohistochemistry, we observed an increase in calretinin expression during aging in surviving OHCs of the cochlea. Similarly, recent studies on cell type–specific transcriptomic analysis on aged mouse cochlear hair cells showed an increase in the expression of calretinin (Calb2 gene) specifically in OHCs at 26 months of age [80]. It has been reported that the calcium-binding protein calretinin plays a critical role in buffering intracellular calcium during cellular stress in all cochlear nuclei [81] and in brain aging [82,83]. In several brain regions like the posteroventral cochlear nucleus (PVCN) and dorsal cochlear nucleus (DCN), there was an increase of calretinin expression in aged mice compared to young animals [84], and this increase in the calcium binding protein was shown to be correlated with OHCs loss [85,86]. Our findings of reduced cholesterol content in aged OHCs due to CYP46A1 increased activity could be another contributing factor to ARHL. In fact, it was shown in aged hippocampal neurons in vitro that increased expression of the CYP46A1 gene is triggered by ROS accumulation [22,35]. Interestingly, since we observed that phytosterols treatment partially restore the OHCs' function, our results open the possibility to explore an innovative therapeutic strategy for preventing or delaying ARHL-presbycusis. Taking into account that C57BL/6 mice are known to lose their hearing capacity early in life, additional work in alternative mice strains, such as CBA/CaJ, which retains their hearing up to older ages, would be necessary to further explore the progression of the decline in cholesterol levels and hearing with aging to prove that this effect is due to age and not also influenced by the overall damage to the OHCs. Altogether, these results are the first proof-of-principle study showing that CYP46A1 activation can lead to hearing deficits due to cholesterol removal from OHCs, with the consequent alterations in the distribution of the motor protein prestin. Moreover, we show that the effect of cholesterol loss in OHCs can be rescued with phytosterols supplementation in the diet.

As stated above, efavirenz treatment leads to a reduction in the cholesterol levels in OHCs that was reflected by an elevation in DPOAEs thresholds. Several studies have reported an increased incidence of auditory dysfunction among HIV/AIDS patients treated with highly active antiretroviral therapy as a side effect [87,88]. Nowadays, the cocktails frequently used in clinical therapies include efavirenz, suggesting that prolonged treatment may cause hearing deficits. However, the mechanisms underlying hearing loss among HIV/AIDS patients treated with antiretroviral therapy are not well understood. Studies in mice indicated a synergistic relationship between anti-HIV drugs and noise, with an increase in DPOAE thresholds consistent with drug-induced damage to OHCs [89]. Efavirenz was also found to decrease the viability of immortalized auditory HEI-OC1 cells in culture in a dose-dependent manner [87]. The

present study reveals for the first time a possible mechanism by which efavirenz treatment can lead to hearing impairment. Moreover, it sheds light into the use of phytosterols as a potential therapeutic and preventive strategy for treating and/or delaying hearing loss in HIV/AIDS patients treated with antiretroviral therapies.

### Limitations of the study and future directions

Our results provide valuable insights into the relationship between cholesterol homeostasis and the physiopathology of the inner ear. However, there are 2 major limitations in this study that should be addressed in future research. The first limitation is the strain used in the study. C57BL/6J mice are widely used for aging studies due to its low aggressivity, which makes them suitable for long-term studies. However, this strain is known to exhibit early-onset hearing loss. Therefore, it is crucial to replicate the same experiments in a different mouse strain without this predisposition. These experiments are fundamental to determine whether the observed effects are specific to C57BL/6J mice or can be generalized to other mouse strains or even humans. Thus, our future research direction will be to explore the effect of aging and efavirenz administration in cholesterol levels in the cochlea using other strains of mice, like CBA/CaJ or BALB/cJ, and to test if phytosterols supplementation can ameliorate aged- and/or antiretroviral-related hearing loss. Second, although our results show that in aged cochlear tissue there is a reduction in cholesterol content in sensory cells of the inner ear—assessed by filipin labeling—and that this reduction is correlated with an increase in the immunofluorescence of CYP46A1, it will be important to test alternative methods for CYP46A1 quantification in specific cell populations. Future studies incorporating techniques such as RNAscope or scRNA-seq are necessary to validate our findings. In conclusion, while our study sheds light on the extent to which alterations in cholesterol homeostasis contributes to the pathophysiology of the inner ear, it is important to consider the aforementioned methodological limitations. Nonetheless, the initial outcomes of our present research offer great promise as they establish the first evidence in favor of phytosterols supplementation as a potential therapeutic strategy in the prevention or treatment of hearing loss.

## Materials and methods

### Ethics statement

All animal care and experimental protocols were performed in accordance with the American Veterinary Medical Association (AVMA) Guidelines for the Euthanasia of Animals (June 2013) as well as Facultad de Medicina, Universidad de Buenos Aires (UBA) Institutional Animal Care and Use Committee (IACUC) guidelines, and best practice procedures. The animal experiments were approved by the Institutional Animal Care and Use Committee of Facultad de Medicina, Universidad de Buenos Aires (UBA) (Approval Number: RESCD-2022-1402-E-UBA-DCT#FMED) and the Institutional Animal Care and Use Committee of Instituto de Investigaciones Biomédicas, Pontificia Universidad Católica Argentina (BIOMED) (Approval Number: 011–2021 and 005–2022).

### Animals

Young (2 months) and old (2 years) C57BL/6J male mice were used in this study. All experimental mice were maintained under specific pathogen-free conditions with a 12-hour light/dark cycle and open access to food and water. The temperature and humidity were set at 22 ± 1˚C and 55% ± 5%, respectively.

## Efavirenz and phytosterols treatment in vivo

The treatment started at 8 weeks of age (2 months), with efavirenz administered in the drinking water at a dose of 0.09 mg/kg/day as described [50]. Efavirenz solutions were replaced every 3 days during the 4 weeks of treatment. Control groups drank water. Efavirenz treatment did not significantly alter the daily consumption of water (Controls: 6.2 ± 0.3 ml, Efavirenz: 6.1 ± 0.2 ml, and Efavirenz plus Phytosterols: 6.3 ± 0.2 ml; one-way ANOVA, df = 2, $p = 0.67$). Phytosterols (Vitatech) were added to standard food pellets (Cooperación, Argentina) at a concentration of 2% (w/w) and administered ad libitum. The phytosterols treatment was performed for 3 weeks (i.e. initiated 1 week after the efavirenz treatment started). At the end of the treatments, mice exhibited similar body weights (Controls: 28.8 ± 0.4 g, Efavirenz: 27.0 ± 0.9 g, and Efavirenz plus Phytosterols: 27.9 ± 0.2 g; one-way ANOVA, d = 2, $p = 0.15$).

## Cochlear function tests

Inner ear physiology, including ABRs and DPOAEs, was performed in mice anesthetized with xylazine (10 mg/kg, IP) and ketamine (100 mg/kg, IP) and placed in a soundproof chamber maintained at 30˚C. Sound stimuli were delivered through a custom acoustic system with 2 dynamic earphones used as sound sources (CDMG15008–03A; CUI) and an electret condenser microphone (FG-23329-PO7; Knowles) coupled to a probe tube to measure sound pressure near the eardrum (for details, see https://www.masseyeandear.org/research/otolaryngology/investigators/laboratories/eaton-peabody-laboratories/epl-engineering-resources/epl-acoustic-system). Digital stimulus generation and response processing were handled by digital I-O boards from National Instruments driven by custom software written in LabVIEW (generously given by Dr. M. Charles Liberman, Eaton-Peabody Laboratories, Massachusetts Eye & Ear Infirmary, Boston, MA). For ABRs, needle electrodes were placed into the skin at the dorsal midline close to the neural crest and pinna with a ground electrode near the tail. ABR potentials were evoked with 5 ms tone pips (0.5 ms rise-fall, with a $\cos^2$ envelope, at 40/s) delivered to the eardrum at log-spaced frequencies from 5.6 to 45.25 kHz. The response was amplified 10,000× with a 0.3- to 3-kHz passband. Sound level was raised in 5 dB steps from 20 to 80 dB sound pressure level (SPL). At each level, 1,024 responses were averaged with stimulus polarity alternated. Threshold for ABR was visually defined as the mean between the lowest stimulus level at which a repeatable peak 1 could be identified in the response waveform and the sound intensity before that. The ABR peak 1 amplitude was computed by offline analysis of the peak to baseline amplitude of stored waveforms. The DPOAEs in response to 2 primary tones of frequency f1 and f2 were recorded at 2f1-f2, with f2/f1 = 1.2, and the f2 level 10 dB lower than the f1 level. Ear canal sound pressure was amplified and digitally sampled at 4 μs intervals. DPOAE threshold was defined as the lowest f2 level in which the signal to noise floor ratio is >1. Offline analysis of ABR thresholds and P1 amplitudes together with DPOAEs thresholds were done by 2 experimenters "blind" to each animal's treatment condition.

## Cochlear processing and immunohistochemistry

Temporal bones were collected from young (control and treated with efavirenz and efavirenz plus phytosterols) as well as from 2-year-old mice. Cochleae were perfused intralabyrinthly with 4% paraformaldehyde (PFA) in phosphate-buffered saline (PBS), postfixed with 4% PFA overnight, and decalcified in 0.12 M EDTA. Cochlear tissues were then microdissected and blocked in 5% normal goat serum with 1% Triton X-100 in PBS for 1 hour, followed by incubation in primary antibodies (diluted in blocking buffer) at 4˚C for 16 hours. The primary antibodies used in this study were as follows: (1) rabbit anti-CYP46A1 antibody (Proteintech

#12486-1-AP, 1:400); (2) mouse anti-calretinin antibody (Millipore, Billerica, MA; MAB1568, 1:1,000); and (3) goat anti-prestin antibody (Santa Cruz Biotechnology Inc. sc22692; 1:700). Tissues were then incubated with the appropriate Alexa Fluor–conjugated fluorescent secondary antibodies (Invitrogen, Carlsbad, CA; 1:1,000 in blocking buffer) for 2 hours at room temperature. Finally, tissues were mounted on microscope slides in FluorSave mounting media (Millipore, Billerica, MA). Confocal $z$-stacks (0.3 μm step size) of the apical, medial, and basal regions from each cochlea were taken using a Leica TCS SPE microscope equipped with 63× (1.5× digital zoom) oil-immersion lens. Image stacks were imported to Fiji software [82] for analysis. Maximum projections were made using the sum of intensities pixel by pixel and the region of interest (ROI) that correspond to each cellular type (IHC, OHC, and SC) were drawn. The mean intensity of CYP46A1 was calculated for each ROI. Pixel intensities of prestin in OHCs were measured along a line drawn through the middle of 1 or 3 consecutives cells in all the cochlear sections from control and treated mice. For the prestin intensity plot as a function of distance (Fig 4B), individual OHCs from the medial region of the cochlea from different sections were analyzed and pixel intensities were plotted along the width of the cell. The average diameter of an OHC is 8 μm. We considered that the intensity values between the region of 0 to 2 μm and 6 to 8 μm (indicated by the pink color of the line in Fig 4B) corresponds to the lateral wall of one individual OHC, and the orange part of the line in Fig 4B indicates the cytoplasm of the cell. We then computed the average fluorescence in the membrane area (Distances 0 to 2 and 6 to 8 μm) and in the cytoplasm (Distances 2 to 6 μm) and calculated the ratio between the membrane and the cytoplasm fluorescence (Fig 4C). For the prestin intensity analysis at all the regions of the cochlea (Fig 4D), OHCs were analyzed in groups of 3 consecutive cells and the mean pixel intensity was plotted as a function of apical, medial, and basal end. OHCs were counted along the whole cochlear sections combining the calretinin positive signal together with light microscopy using Nomarski optics, based on the assessment of all present and absent hair cells in all the sections.

## Filipin labeling of whole mounts organ of Corti

The sensory epithelium was isolated and washed twice with PBS, fixed with 4% PFA for 30 minutes, and stained with the fluorescent molecule filipin (SIGMA, 125 μg/ml) and AlexaFluor 647 phalloidin (Invitrogen, Carlsbad, CA; 1:20) for 60 minutes in the darkness. The samples were then washed twice with PBS and mounted on microscope slides in FluorSave mounting media. Confocal z-stacks (0.3 μm step size) of the apical, medial, and basal regions from each cochlea were taken using the Leica TCS SPE microscope equipped with 63× (1.5× digital zoom) oil-immersion lens. Image stacks were examined using Fiji software [90]. Filipin pixel intensity was measured by drawing a line through the middle of 1 or 2 consecutive cells and plotted as either pixel intensities or relative intensity. All the OHCs were analyzed from each confocal z-stack [45].

## Statistical analysis

Data are presented as group means ± standard error (SEM) and were analyzed with R Statistical Software [91]. Kruskal–Wallis nonparametric ANOVA followed by Dunn's post-tests and Mann–Whitney tests were used to determine statistical significance in cases where data were nonnormally distributed (peak 1 amplitude between groups, ABR and DPOAE thresholds). For 3-group comparisons of data normally distributed, one-way ANOVA followed by Tukey's post-tests were used to determine statistical significance (filipin and prestin intensities). For 2-group comparisons, a $t$ test was performed (CYP46A1 intensity, %OHCs and filipin intensity). A $p < 0.05$ was considered statistically significant at a 95% confidence level.

## Supporting information

**S1 Fig. Calretinin immunostaining in young and aged mouse cochlea.** (**A**) Representative 10× low-magnification confocal images in the xy projection of whole mounts organ of Corti from the apical/medial region immunolabeled for calretinin from young (left; $n = 5$) and old (right; $n = 6$) C57BL/6J mice (scale bar = 200 μm). Insets: 40× magnification images of the selected area indicated by a white rectangle (scale bar = 50 μm). Arrows point to OHCs and arrowheads point to lost IHCs in aged ear tissue. (**B**) Bar graph showing the percentage of OHCs survival in aged mice compared to young animals. At 2 years of age, there was a 70% of OHCs death along the whole length of the cochlea. Asterisks represent the statistical significance ($t$ test, **** = $p < 0.0001$). The data underlying this figure can be found at https://osf.io/xpemk/.
(TIF)

**S2 Fig. Cholesterol content in IHCs and SCs from control, efavirenz, and efavirenz plus phytosterols-treated mice.** (**A**) Representative confocal images of whole mounts organ of Corti from the medial region stained with filipin in control, treated with efavirenz, and treated with efavirenz plus Phy mice. (Scale bar = 30 μm). Pink lines indicate the diameter of IHCs and SCs as an example on how we select cells to measure the pixel intensity. (**B**) Relative intensity bar graph showing the changes in filipin staining in IHCs in the 3 groups of mice at the 3 regions of the cochlea (apical, medial, and basal) (Control: $n = 48$ IHCs at the apical, 128 IHCs at the medial, and 66 IHCs at the basal region from 6 mice; efavirenz: $n = 81$ IHCs at the apical, 75 IHCs at the medial, and 35 IHCs at the basal region from 6 mice and efavirenz together with phytosterols: $n = 93$ IHCs at the apical, 114 IHCs at the medial, and 60 IHCs at the basal region from 6 mice). (**C**) Relative intensity bar graph showing the changes in filipin staining in SCs in the 3 groups of mice at the 3 regions of the cochlea (Control: $n = 10$ SCs at the apical, 21 SCs at the medial, and 10 SCs at the basal region; efavirenz: $n = 22$ SCs at the apical, 12 SCs at the medial, and 11 SCs at the basal region and efavirenz together with phytosterols: $n = 14$ SCs at the apical, 11 SCs at the medial, and 10 SCs at the basal region). There was a significant reduction in cholesterol levels after efavirenz treatment in both IHCs and SCs. Treatment with efavirenz together with Phy showed an increase in filipin staining at the 3 regions compared with mice treated with efavirenz alone, except in IHCs at the apical region. However, when compared to controls, there was a significant reduction in filipin staining at the 3 regions of the cochlea in both IHCs and SCs. Group means ± SEM are shown. Asterisks represent the statistical significance (one-way ANOVA, followed by Tukey's test, = *** $p < 0.001$, **** = $p < 0.0001$). The data underlying this figure can be found at https://osf.io/xpemk/.
(TIF)

## Acknowledgments

We thank Dr. Ana Belén Elgoyhen for her insightful discussions and continuous support, and Dr. Carina Porporatto for providing reagents.

## Author Contributions

**Conceptualization:** Mauricio G. Martin, María Eugenia Gomez-Casati.

**Data curation:** Alejandro O. Sodero, Valeria C. Castagna, Mauricio G. Martin, María Eugenia Gomez-Casati.

**Formal analysis:** Alejandro O. Sodero, Valeria C. Castagna, Setiembre D. Elorza, Sara M. Gonzalez-Rodulfo, Jimena A. Ballestero, Mauricio G. Martin, María Eugenia Gomez-Casati.

**Funding acquisition:** Mauricio G. Martin, María Eugenia Gomez-Casati.

**Investigation:** Alejandro O. Sodero, Valeria C. Castagna, Jimena A. Ballestero, Mauricio G. Martin, María Eugenia Gomez-Casati.

**Methodology:** Alejandro O. Sodero, Valeria C. Castagna, María A. Paulazo, Mauricio G. Martin, María Eugenia Gomez-Casati.

**Project administration:** Mauricio G. Martin, María Eugenia Gomez-Casati.

**Resources:** Mauricio G. Martin, María Eugenia Gomez-Casati.

**Software:** Mauricio G. Martin, María Eugenia Gomez-Casati.

**Supervision:** Alejandro O. Sodero, Mauricio G. Martin, María Eugenia Gomez-Casati.

**Validation:** Alejandro O. Sodero, Valeria C. Castagna, Mauricio G. Martin, María Eugenia Gomez-Casati.

**Visualization:** Alejandro O. Sodero, Valeria C. Castagna, Mauricio G. Martin, María Eugenia Gomez-Casati.

**Writing – original draft:** Alejandro O. Sodero, Valeria C. Castagna, Mauricio G. Martin, María Eugenia Gomez-Casati.

**Writing – review & editing:** Valeria C. Castagna, Jimena A. Ballestero, Mauricio G. Martin, María Eugenia Gomez-Casati.

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
