## [Editor Report · Decision Letter 0]

15 Sep 2022

Dear Dr Gomez-Casati, 

Thank you for submitting your manuscript entitled "Phytosterols reverse hearing impairments caused by antiretroviral treatment. Implications in presbycusis." for consideration as a Research Article by PLOS Biology.

Your manuscript has now been evaluated by the PLOS Biology editorial staff, as well as by an academic editor with relevant expertise, and I am writing to let you know that we would like to send your submission out for external peer review. I should note that, while we find the topic of your study interesting, we are yet to make a firm call about whether the study provides enough cellular mechanistic insights for our Research Article format. We will be looking for some enthusiasm from the reviewers.

Before we can send your manuscript to reviewers, we need you to complete your submission by providing the metadata that is required for full assessment. To this end, please login to Editorial Manager where you will find the paper in the 'Submissions Needing Revisions' folder on your homepage. Please click 'Revise Submission' from the Action Links and complete all additional questions in the submission questionnaire.

Once your full submission is complete, your paper will undergo a series of checks in preparation for peer review. After your manuscript has passed the checks it will be sent out for review. To provide the metadata for your submission, please Login to Editorial Manager (https://www.editorialmanager.com/pbiology) within two working days, i.e. by Sep 19 2022 11:59PM.

Kind regards,

Lucas

Lucas Smith, Ph.D.

Associate Editor

PLOS Biology

lsmith@plos.org

---

## [Decision Letter · Decision Letter 1]

7 Nov 2022

Dear Dr Gomez-Casati,

Thank you for your patience while your manuscript "Phytosterols reverse hearing impairments caused by antiretroviral treatment. Implications in presbycusis" was peer-reviewed at PLOS Biology. Your manuscript has been evaluated by the PLOS Biology editors, an Academic Editor with relevant expertise, and by several independent reviewers.

As you will see in the reviewer reports, which can be found at the end of this email, although the reviewers find the work potentially interesting, they have also raised a number of important concerns, which would need to be thoroughly addressed before we can consider your manuscript further for publication. Having discussed the reviews with the Academic Editor, it is clear that a substantial amount of work would be required to expand and strengthen the mechanism provided here, which we think would be essential in order for this study to meet the criteria for publication in PLOS Biology.

However, given our and the reviewer interest in your study, we would be open to inviting a comprehensive revision of the study that thoroughly addresses all the reviewers' comments, with new analyses and data. To this end, we are willing to relax our standard revision time to allow you 6 months to revise your study. Please email us (plosbiology@plos.org) if you have any questions or concerns, or envision needing a (short) extension.

Given the extent of revision that would be needed we cannot make a decision about publication until we have seen the revised manuscript and your response to the reviewers' comments. Your revised manuscript would likely need to be seen by the reviewers again.

**IMPORTANT - SUBMITTING YOUR REVISION**

*Resubmission Checklist*

*Published Peer Review*

*PLOS Data Policy*

*Blot and Gel Data Policy*

Sincerely,

Lucas

Lucas Smith, Ph.D.

Associate Editor

PLOS Biology

lsmith@plos.org

REVIEWS:

Reviewer #1: In the manuscript 'Phytosterols reverse hearing impairments caused by antiretroviral treatment. Implications in presbycusis.' Sodero et al., describe an increase in CYP46A1 leads to a decrease of cholesterol in the outer hair cells of aged mice and mice treated with an antiretroviral drug used in HIV/AIDS treatment. They are able to show that administrating this drug in young mice affects outer hair cell function as measured via DPOAEs and can be partially rescued by feeding phytosterols to the same animals. In general, I think this is a very interesting study that is of interest for hearing research. However, I do have some concerns with this study.

1) Fig1A: The calretinin stain is very hard to see in the OHC in the young group, but very clear in the aged group. Since it is reported (e.g. Dechesne CJ, Rabejac D, Desmadryl G. Development of calretinin immunoreactivity in the mouse inner ear. J Comp Neurol. 1994 Aug 22;346(4):517-29. doi: 10.1002/cne.903460405. PMID: 7983242.) that calretinin is not expressed in the outer hair cells in the adult (>P22) that is very surprising to me. Are you sure those are OHCs?

2) The animal numbers are not completely clear to me. Did you use the same animals for the different immunostainings? Why were there only 3 animals for the filipin staining in the young group but 5 for the CYP46A1 staining in the same age group?

3) While the authors do mention that C57Cl/6 mice are known to lose their hearing early it would be beneficial to discuss why they chose this mouse strain anyways vs a mouse strain like CBA/CaJ which are known to retain their hearing up to an old age. In that case the conclusion of age being the driving factor of the reduction and redistribution of cholesterol in the OHCs would have had much more merit. In the present study it is hard to prove that this effect is due to age and not also influence by the overall damage to the OHCs. This should be discussed

4) In the aging study it is mentioned that an increase in CYP46A1 and decrease in filipin staining can be observed not only in OHCs but also IHCs and SCs. Was that also the case with the drug treatment?

5) Fig 1A: the merged of the aged animals does not line up with the Calretinin only stain. 

6) Fig 4A: in the Efavirenz+Phyto group ot seems as if Filipin is located to the cell membrane vs in the other groups where it is more central. Also, the Phalloidin intensity is reduced in this group compared to the others. And in the Efavirez group it seems to be more diffusely distributed. Please explain this.

7) I am not sure about the reasoning of using the average of either two or three OHCs for the analysis in Figures 4 & 5. 

8) Is there a reason why you represent different areas in the cochlea for your filipin vs prestin stain intensity?

9) How did you select the two/three OHCs to measure across?

Minor Comments:

1) Please use the same citation format throughout the manuscripts.

2) In the table with the questions from the journal you state that an ethics approval was not necessary for this study despite conducting an animal study (which was listed as one of the reasons to include an ethics statement).

3) It is not always clear if the pink arrow heads are considered part of the line or if it is only the distance between the two arrow heads that is marked.

4) In the results section you write that you treat for 3 weeks with the phytosterols. In the methods section it is stated that the treatment was 4 weeks. Please correct where necessary.

Reviewer #2: Sodero et al. studied the role of cholesterol in hearing loss by analyzing the relationship between the decrease in cholesterol levels and outer hair cell degeneration. They detected elevation of DPOEA thresholds after administration of the mice with efavirenz, an antiretroviral which pharmacologically activates an enzyme responsible for cholesterol turnover (CYP46A1). Interestingly, the elevation in DPOAE thresholds due to efavirenz administration was prevented by dietary complementation with phytosterols. If this observation holds true, it would be of high interest as a strategy for preventing hearing loss due to age-related issues and ototoxic drugs. However, there are some important concerns about these claims since the experiments do not provide a clear mechanism of action. Also, there is no elevation of auditory thresholds (ABRs) caused by the drug administration, which dampens enthusiasm for work. 

Some important points to consider:

- Can the authors use qRT-PCR for quantifying CYP46A1 levels? Immunostaining can be highly variable across experiments. Is the difference in CYP46A1 quantity between young and old mice stronger in the base or the apex? Only mid sections were used in Fig 1. 

- Could you explain how were the ROIs defined for the quantification of CYP46A1 in the supporting cells? 

- In Fig 2, since there is substantial hair cell loss in aged mice, the orientation of the cells is less consistent which makes it difficult to compare with younger mice that have a better-preserved Organ of Corti. How was the IHC cholesterol concentration quantification performed? In which part of the cell body was the analysis done?

- Is the effect of efavirenz (and phytosterols) dose dependent? Could higher doses of the drug cause ABRs elevations? The fact that the DPOEAs thresholds are elevated with no effect on ABRs is surprising because OHC function is necessary to maintain normal ABR levels, and also this indicates that there is no evident hearing loss associated with efavirenz administration. 

- Could the authors perform ABR recordings of the treated mice at 2 months of age prior to the drug treatments? Then, comparisons can be made within each animal.

- Line 407-409: "Studies in mice suggested a synergistic relationship between anti-HIV drugs and noise, with an increase in DPOAE thresholds consistent with drug-induced damage to OHCs" Would the efavirenz-treated animals be more sensitive to noise-exposure? 

- Can you show the filipin staining of IHC after efavirenz (and Phy) treatment (Fig 4)? 

- Are the newly delivered phytosterols being incorporated into the membrane? Can you radiolabel the molecules and do a quantitative analysis of these lipids incorporation? 

- Lines 372-373: "…with an apparent recovery of the normal prestin distribution in the lateral wall." Can you show the quantification of prestin intensity in the periphery of the cell? 

- Why were only male mice used in the study?

Some other minor points:

- Could you enhance the contrast/brightness of calretinin staining in Fig 1?

- Could you also include ABR and DPOAE waveforms? 

- Given that efavirenz produces dyslipidemia but also dysglycemia (doi: 10.1097/MD.0000000000002385), how do you know its effect is not due to the alteration on glucose metabolism?

- Does the treatment with efavirenz change the level of CYP46A1 levels in the cochlea? 

Reviewer #3: Numerous studies have shown that cholesterol imbalances in the development of neurodegenerative diseases, but the role of cholesterol in the physiopathology of the inner ear has not been studied. This paper characterized cholesterol homeostasis in the inner ear of young and old mice, and tested the hypothesis that cholesterol deficiency might play a role in cochlear pathologies and hearing loss by analyzing CYP46A1 levels and cholesterol content in the inner ear of young and aged mice. The authors also tested the effects of efavirenz on the auditory function. The effect of dietary supplementation with phytosterols was also tested in the inner ear of mice treated with efavirenz. Overall, their findings are new and interesting, but no histological examinations of cochlear cells/microstructures (e.g., hair cell counts, spiral ganglion neuron/auditory nerve fiber density, stria vascularis thickness, ribbon synapse counts, etc.) was performed, although the aim of the study was to test the hypothesis that cholesterol deficiency might play a role in cochlear pathologies and hearing loss. Cholesterol levels were not measured in the whole inner ear tissues from the experimental groups (young, old, control, efavirenz, and efavirenz + phytosterol groups). In addition, I have some questions about the age of the old group and the statistical method. I elaborate on these and other points below. 

Line 94. The authors stated that the aim of this study was to test the hypothesis that cholesterol deficiency might play a role in cochlear pathologies and hearing loss, but no histological examination of cochlear cells/microstructures (e.g., hair cell counts, spiral ganglion neuron/auditory nerve fiber density, stria vascularis thickness, ribbon synapse counts, etc.) was performed in the young, old, control, efavirenz, and efavirenz + phytosterol groups. So I don't think the proposed hypothesis was thoroughly tested in the current study, and the overall finding seems inconclusive in terms of testing the hypothesis. 

Method. I understand cholesterol levels were measured using the cholesterol-binding fluorescent antibiotic filipin in the fixed hair cells, but I do think total cholesterol levels should also be measured in the whole inner ear tissues of all the experimental groups (young, old, control, efavirenz, and efavirenz + phytosterol groups) to test the hypothesis that cholesterol deficiency might play a role in cochlear pathologies and hearing loss. 

Line 109. C57BL/6J mice develop early onset severe and progressive hearing loss by 12-15 months of age, with the disruption of both outer and inner hair cells, due to the Cdh23Ahl allele (https://www.jax.org/strain/000664), so it is not clear to me why 24 months old mice were used when they were near deaf and likely lost most of the hair cells. I think use of 3 age groups (e.g., 2, 6, 12 months old groups) or 2 age groups (e.g., 2, 12 months old groups instead of 2 and 24 months old) would make more sense for the measurements of CYP46A1 and filipin labeling.

Fig. 3/Fig. 8. I am not sure how this can be Kruskal-Wallis since (I think) the factors are 3 experimental groups and frequencies. My understanding is that Kruskal-Wallis cannot handle two way ANOVA data in the non-parametric sense. Could you clarify this? 

In addition, is there any reason 6 of each group was used for the ABR and DPOAE tests?

Line 158. The aim of this experiment was to test the hypothesis that CYP46A1 activation might reduce the cholesterol content in young mice, leading to hearing deficits. So, I think histological examination of cochlear cells (such as hair cell counts) should also be performed at the end of the treatment (at 3 months of age) to validate the ABR test results. In addition, what is the rationale for starting efavirenz treatment at 2 months of age for just 4 weeks as opposed to starting at a later time point (e.g., 5 months of age) for a longer period (e.g., 3 months)?

Lines 231 and 265. Again, what is the rationale for starting efavirenz + phytosterol treatment at 2 months of age for just 3 weeks as opposed to starting at a later time point (e.g., 5 months of age) for a longer period (e.g., 3 months)?

Fig. 4, 5, 9, 10 (bar graphs). I think error bars should be added to the bar graphs.

---

## [Decision Letter · Decision Letter 2]

30 Jun 2023

Dear Dr Gomez-Casati,

Thank you for your patience while we considered your revised manuscript "Phytosterols reverse hearing impairments caused by antiretroviral treatment. Implications in presbycusis" for consideration as a Research Article at PLOS Biology. Your revised study has now been evaluated by the PLOS Biology editors, the Academic Editor and the original reviewers.

The reviewers largely agree that the manuscript has been strengthened in the revision, however Reviewers 1 and 2 have a number of lingering concerns and additional comments that we think should be addressed. In some cases, Reviewer 2 has requested additional data. While adding this data would strengthen the study, after discussion with the Academic Editor, we think this would not strictly be required for publication at PLOS Biology. However, in the absence of new data, you should address Reviewer 2's concerns by adding a section to the discussion outlining some of the methodological considerations and limitations of the study. 

In addition to addressing the reviewer comments, below, we also ask that you please address the following editorial requests: 

1) TITLE: We think that the title should be edited slightly to avoid punctuation and for clarity. If you agree, we suggest you change it to something like "Phytosterols reverse age-related cochlear changes and hearing impairments caused by antiretroviral treatment in mice"

2) ARTICLE TYPE: After discussion within the team, we think that your manuscript is closest in scope to our 'Short Report' article type, and so we request that you change it accordingly. Short Reports can have a maximum of 4 figures and so this will require that you condense 2 of your figures or move one to the supplement. We suggest that Figures 1 and 2 might be combined to reduce the figure number to 4. For more information about PLOS Biology Short Reports, see here: https://journals.plos.org/plosbiology/s/what-we-publish#loc-short-reports

3) DATA: Thank you for providing the underlying data for your study as a deposition on OSF. Can you please add a sentence to each figure legend (including supplemental) referencing this dataset? For example, you can add the sentence "the data underlying this figure can be found at https://osf.io/xpemk/"

We are pleased to offer you the opportunity to address the remaining points from the reviewers and the abovementioned editorial requests in a revision that we anticipate should not take you very long. We will then assess your revised manuscript and your response to the reviewers' comments with our Academic Editor aiming to avoid further rounds of peer-review, although might need to consult with the reviewers, depending on the nature of the revisions.

**IMPORTANT - SUBMITTING YOUR REVISION**

*Resubmission Checklist*

*Published Peer Review*

*PLOS Data Policy*

*Blot and Gel Data Policy*

Sincerely,

Luke

Lucas Smith, Ph.D.

Senior Editor

PLOS Biology

lsmith@plos.org

REVIEWS:

Reviewer #1, Alice L. Burghard (note, reviewer 1 has signed this review): In the revised version of the manuscript "Phytosterols reverse hearing impairments caused by antiretroviral treatment. Implications in presbycusis." the authors have addressed most of my previous comments. 

However, I still have a few comments that should be addressed.

1) You report group differences by stating the significance found in the ANOVA, please also include the results from the post-hoc test. 

2) I am not sure what happened, but all the figures are blurry. The supplemental material is not blurry. I would recommend that you use that format throughout the manuscript.

3) In Fig 3 you give a scale bar of 50 dB. Please give the reference if that is dB SPL or something else.

4) When you plot distance and the pink dotted line (e.g. Fig 4B) it is not clear immediately that the pink dashed line is representing the dashed line from panel A. Maybe instead of assigning it a level on the y-axis put it underneath the x-axis? That might also explain the 'distance' measurement a little better. Or maybe use the term width? In the figure caption the term length is used. Please make sure you keep the terms consistent throughout the manuscript. 

5) Also in Fig 4, please explain the pink dashed lines already for A and not only in B.

6) Page 26 line 645 and 646: I think you mean 0-2 and 6-8 and 2-6 not 2-4.

Reviewer #2: In the revised version of the manuscript, Sodero et al. did not provide alternative methods for CYP46A1 quantification, so I am still not convinced that CYP46A1 increased expression is one of the mechanisms behind ARHL. Also, quantification of cholesterol using filipin staining in an intact (young) and a degenerated (aged) tissue makes comparisons between groups unreliable (I will elaborate on that). As the authors mentioned, they are exploring the effect of aging and efavirenz administration in cholesterol levels in the cochlea using other strains of mice, which would make the results stronger. Determination of a dose-response curve for the effect of efavirenz in the cochlea will be fundamental to understand the effect this drug has on patients. I do believe that the efavirenz results are very interesting since they show that localization of prestin in OHCs is affected by the drug administration and reverted by phytosterols consumption, which could explain the DPOAEs thresholds elevations and recovery respectively.

Some points to work on are:

* Figure 1: According to the immunostaning data, the CYP46A1 intensity level is very different in the organ of Corti of aged vs young mice and this is not just in hair cells so the fact that it was not detected through qPCR makes the staining data less reliable, even though the ages were not exactly matched with the ones shown in the paper. The authors should perform RNA Scope (or other quantitative in situ hybridization method) to resolve the discrepancy in an age where they detect the decrease in protein level in a cell-specific manner.

* Figure 2: In the case of young animals, OHC are oriented more perpendicular to the plane of the photo, but this is not the case for aged animals since there is substantial hair cell death and the orientation of the cells becomes less defined. This makes comparisons between groups very inconsistent because the cell volume that is being included in the quantification zone is larger for the young group than for the aged group. 

* Figure 3: How did the authors call the ABR thresholds? If manually done, was it done by more than one person? Was this done blindly? An algorithm can be used for that (DOI: 10.1016/j.heares.2019.107782). In the efavirenz + phytosterols traces, it looks like another person or an algorithm could very likely pick the intensity immediately bellow the one called by the authors since there appears to be small but clear peaks 1 & 2.

I agree with lines 540-545 where the authors mention that it is relevant to study this phenomenon in other mouse strains in order to confirm that the effect seen is not related to the early HL that the C57BL/6 line presents. In that sense, I believe that including an alternative method for CYP46A1 quantification will be important to make the immunostaining results more reliable, together with an antibody that can label OHC homogeneously across ages (like Myo7a).

Reviewer #3: The authors have addressed all of my major concerns.

---

## [Editor Report · Decision Letter 3]

18 Jul 2023

Dear Dr Gomez-Casati,

Thank you for the submission of your revised Short Report "Phytosterols reversal of antiretroviral-induced hearing loss and its implications in presbycusis" for publication in PLOS Biology, and thank you for addressing the reviewer and editorial requests in your revision. On behalf of my colleagues and the Academic Editor, Manuel S. Malmierca, I am pleased to say that we can in principle accept your manuscript for publication, provided you address any remaining formatting and reporting issues. These will be detailed in an email you should receive within 2-3 business days from our colleagues in the journal operations team; no action is required from you until then. Please note that we will not be able to formally accept your manuscript and schedule it for publication until you have completed any requested changes.

PRESS

Sincerely, 

Lucas Smith, Ph.D.

Senior Editor

PLOS Biology

lsmith@plos.org